# BEAMRAD: A tool for Creating and Assessing Medical Dataset Documentation

**Maria Galanty**[*,1,2] (iD)                M.GALANTY@UVA.NL

**Dieuwertje Luitse**[*,1] (iD)                D.LUITSE@UVA.NL

**Alexander P. Vlaar**[2] (iD)          A.P.VLAAR@AMSTERDAMUMC.NL

**Clara I. Sánchez**[1,2] (iD)        C.I.SANCHEZGUTIERREZ@UVA.NL

**Tobias Blanke**[1] (iD)                  T.BLANKE@UVA.NL

**Ivana Išgum**[1,2] (iD)            I.ISGUM@AMSTERDAMUMC.NL

[1] *University of Amsterdam, The Netherlands*

[2] *Amsterdam UMC, University of Amsterdam, The Netherlands*

**Editors:** Accepted for publication at MIDL 2025

## Abstract

Medical datasets drive deep learning in medical imaging but may introduce biases that impact model performance and clinical applicability. To address these bias challenges, we introduce **BEAMRAD**, a dynamic tool to create and assess medical dataset documentation. BEAMRAD systematically evaluates documentation, and links insufficient reporting to potential biases. Through an exemplary assessment of publicly available medical datasets, we highlight gaps in the evaluated dataset documentation, including inconsistencies in data annotation, error quantification, and dataset limitations reporting. We propose to address these issues with three key improvements: stricter repository oversight, reflective documentation practices, and adaptable documentation.

**Keywords:** Medical datasets, Dataset documentation, Bias

## 1. Introduction

High-quality public datasets are vital to advance deep learning (DL) models and applications in medical imaging and healthcare more broadly. However, these datasets often introduce biases that can impact model performance, reliability, and robustness, potentially affecting their clinical applicability (Santa Cruz et al., 2021; Mehrabi et al., 2021). To detect and mitigate such bias, scholars have increasingly invested in the development of standardized dataset documentation guidelines (Gebru et al., 2021; Moons et al., 2014; Maier-Hein et al., 2020; Rostamzadeh et al., 2022), which are recognized as valuable resources to enhance data quality. However, due to their broader application, they often do not specifically focus on the dataset documentation practices necessary to identify and address bias origins in DL-based medical imaging. We, therefore, present and demonstrate the **B**ias **E**valuation **A**nd **M**onitoring for Transparent and **R**eliable Medic**a**l **D**atasets (**BEAMRAD**) tool (Galanty et al., 2024). This focused, yet comprehensive dynamic tool supports the creation of documentation for new datasets and the evaluation of existing dataset documentation. Such qualitative assessments are important, as there is limited research on how well current documentation guidelines are being followed.

---

[*] Contributed equally

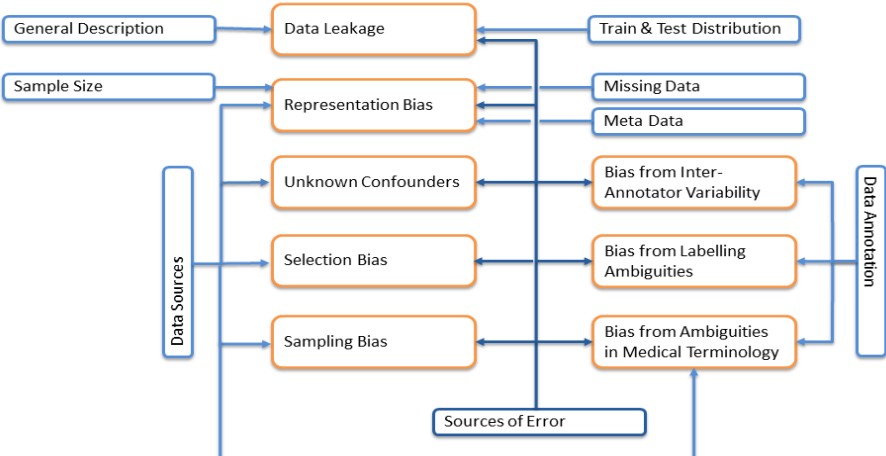

Figure 1: Overview of the potential bias implications for individual sections of the BEAM-
RAD tool. The arrows connect BEAMRAD sections (blue) with potential biases
(orange) that may arise and become difficult to detect without clear documenta-
tion of specific dataset elements.

## 2. Methods

Building on existing guidelines (Gebru et al., 2021; Moons et al., 2014; Maier-Hein et al.,
2020) and in consultations with a multidisciplinary research consortium in Health AI[1], we
have created **BEAMRAD** tool (Galanty et al., 2024). We have linked its categories to
potential biases that may arise from insufficient dataset documentation (see Figure 1). By
establishing these connections, BEAMRAD not only facilitates dataset transparency but
also aids in identifying and mitigating biases at an early stage, ensuring its applicability in
medical AI research.

   To demonstrate and evaluate the applicability of BEAMRAD in practice, we applied
the tool to 37 publicly available medical image and signal datasets, including Magnetic Res-
onance Imaging (MRI), Color Fundus Photography (CFP), and Electrocardiogram (ECG)
datasets (Galanty et al., 2024). The assessment covered key documentation elements, in-
cluding title, general description, data usage, data sources, metadata, sample size, missing
data, training and test set distribution, data annotation, data preprocessing, and sources of
error. This analysis provided critical insights into current dataset documentation practices
and highlighted areas requiring improvement.

## 3. Results

Our analysis reveals notable inconsistencies in the dataset documentation. While certain
sections—like data usage, sample size, and preprocessing—are well-covered in over 75% of

---

1. Research consortium of AI researchers working in medical domain, expert researchers in Critical Data
   Studies and AI ethics as well as Health Law

datasets, other categories are partially well-documented or entirely under-documented. The *data sources* category shows wide variation. Most datasets mention data origin, but fewer than half report acquisition dates, which might be important when considering confounders, e.g. the pre- and post-pandemic context. The availability of inclusion criteria information also varies per modality, with CFP datasets reporting this in 80% of cases, compared to just 35% for ECG. Annotation practices, essential for supervised learning and evaluation, are especially under-documented. While many datasets describe annotation methods, only a subset (26%) includes detailed annotators instruction, and only 23% provide information on annotator selection and training. Furthermore, potential sources of error in datasets—such as those introduced through data annotations—are not well-reported, with only 25% of datasets acknowledging them and just 13% attempting to quantify these errors. Our analysis shows that only a small fraction of datasets explicitly address these issues, with MRI datasets offering the most insight into the significant error sources in annotations. Although some datasets demonstrate good practices, like using overlap measures to calculate annotation variability or implementing periodic annotator monitoring, these cases are the exception rather than the norm.

Even though datasets are documented in inconsistent ways, we do not find a significant difference in the amount of documented information for image datasets (MRI and CFP) and signal datasets (ECG). However, when comparing the results related to data annotation for CFP and MRI datasets from the same platform, it becomes evident that CFP datasets provide more comprehensive documentation. This may suggest that the adherence to and interpretation of existing documentation guidelines by individual dataset creators is even more important to take into account in this dynamic process.

## 4. Conclusions

Inconsistent documentation of medical datasets undermines transparency, bias mitigation, and AI reliability. Addressing these issues requires three key improvements: **stricter repository oversight**, **reflective documentation practices**, and **adaptable documentation**. First, platform repositories should enforce stronger validation—e.g., PhysioNet's metadata checks—to ensure documentation quality and consistency across datasets (Jiménez-Sánchez et al., 2024). Second, documentation must go beyond checkbox compliance; creators should critically reflect on what information is needed to mitigate biases and enhance interpretability. Third, as datasets evolve (e.g., through new annotations or subsets), documentation should be updated accordingly—revising only the modified sections to keep the documentation clear and concise. To support this effort, we developed a GitHub repository [2] featuring two versions of BEAMRAD: one for guiding dataset documentation and another for evaluating existing documentation. The tool serves both data creators and users, and welcomes community input and contributions.

## Acknowledgments

This work was supported by the University of Amsterdam Research Priority Area Artificial Intelligence for Health Decision-making

---

2. https://github.com/MariaGalanty/BEAMRAD

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
