# OpenReview forum: "BEAMRAD: A tool for Creating and Assessing Medical Dataset Documentation"
_MIDL.io/2025/Short_Papers — MIDL 2025 - Short Papers_

### Official Review · Reviewer_Pdv5 · 2025-04-23

**Rating:** 4
**Confidence:** 5

**Summary:**

The Bias Evaluation And Monitoring for Transparent and Reliable Medical Datasets (BEAMRAD) is a tool that systematically evaluates documentation and identifies how insufficient reporting can lead to potential biases. By examining publicly available medical datasets, the authors highlight gaps in documentation, including inconsistencies in data annotation and a lack of error quantification. The authors propose three key improvements to address these issues: stricter repository oversight, reflective documentation practices, and adaptable documentation standards.

**Strengths:**

- The manuscript is well written and structured.
- The tool focuses on the qualitative assessment of dataset documentation, an aspect that is often overlooked.
- The authors provide two documents to guide dataset documentation creation and to evaluate existing documentation.
- Their analysis reveal inconsistencies in the documentation of the analyzed datasets.

**Weaknesses:**

- The quantification in the Section 3. Results —such as reporting that “75% of datasets” or “80% of cases” exhibit certain characteristics—is not clearly explained. I suggest clarifying how the qualitative analysis was translated into these quantitative figures.
- In addition to discussing “Datasheets for datasets” it would be relevant to also consider "Healthsheet" [1].
- MRI datasets may contain fewer errors due to the existence of standards for preprocessing and dataset documentation, such as the Brain Imaging Data Structure (BIDS) [2].
- I would suggest to host a Markdown file in the GitHub repository that can be read directly in the browser, eliminating the need to download a DOCX file.

References:

[1] Rostamzadeh, N., Mincu, D., Roy, S., Smart, A., Wilcox, L., Pushkarna, M., ... & Heller, K. (2022, June). Healthsheet: development of a transparency artifact for health datasets. In *Proceedings of the 2022 ACM Conference on Fairness, Accountability, and Transparency* (pp. 1943-1961).

[2] Gorgolewski, K. J., Auer, T., Calhoun, V. D., Craddock, R. C., Das, S., Duff, E. P., ... & Poldrack, R. A. (2016). The brain imaging data structure, a format for organizing and describing outputs of neuroimaging experiments. *Scientific data*, *3*(1), 1-9.

---

### Decision · Program_Chairs · 2025-05-01

Accept